# Evaluate the Toxicity of Pyrethroid Insecticide Cypermethrin before and after Biodegradation by *Lysinibacillus cresolivuorans* Strain HIS7

**DOI:** 10.3390/plants10091903

**Published:** 2021-09-14

**Authors:** Ebrahim Saied, Amr Fouda, Ahmed M. Alemam, Mahmoud H. Sultan, Mohammed G. Barghoth, Ahmed A. Radwan, Salha G. Desouky, Islam H. El Azab, Nihal El Nahhas, Saad El-Din Hassan

**Affiliations:** 1Department of Botany and Microbiology, Faculty of Science, Al-Azhar University, Nasr City, Cairo 11884, Egypt; hema_almassry2000@azhar.edu.eg (E.S.); ahmed.alemam@azhar.edu.eg (A.M.A.); prof.mahmoud@azhar.edu.eg (M.H.S.); mohamed_gamal.sci@azhar.edu.eg (M.G.B.); ahmedradwan@azhar.edu.eg (A.A.R.); 2Botany and Microbiology Department, Faculty of Science, Suez University, Suez 41522, Egypt; abo_bara2w@yahoo.com; 3Food Science & Nutrition Department, College of Science, Taif University, P.O. Box 11099, Taif 21944, Saudi Arabia; i.helmy@tu.edu; 4Department of Botany and Microbiology, Faculty of Science, Alexandria University, Alexandria 21526, Egypt; nihal.elnahhas@alexu.edu.eg

**Keywords:** biodegradation, a pyrethroid insecticide, cypermethrin, *Lysinibacillus* sp., toxicity, seed germination, in-vitro cytotoxicity

## Abstract

Herein, bacterial isolate HIS7 was obtained from contaminated soil and exhibited high efficacy to degrade pyrethroid insecticide cypermethrin. The HIS7 isolate was identified as *Lysinibacillus cresolivuorans* based on its morphology and physiology characteristics as well as sequencing of 16S rRNA. The biodegradation percentages of 2500 ppm cypermethrin increased from 57.7% to 86.9% after optimizing the environmental factors at incubation condition (static), incubation period (8-days), temperature (35 °C), pH (7), inoculum volume (3%), and the addition of extra-carbon (glucose) and nitrogen source (NH_4_Cl_2_). In soil, *L. cresolivuorans* HIS7 exhibited a high potential to degrade cypermethrin, where the degradation percentage increased from 54.7 to 93.1% after 7 to 42 days, respectively. The qualitative analysis showed that the bacterial degradation of cypermethrin in the soil was time-dependent. The High-Performance Liquid Chromatography (HPLC) analysis of the soil extract showed one peak for control at retention time (R.T.) of 3.460 min and appeared three peaks after bacterial degradation at retention time (R.T.) of 2.510, 2.878, and 3.230 min. The Gas chromatography–mass spectrometry (GC–MS) analysis confirmed the successful degradation of cypermethrin by *L. cresolivuorans* in the soil. The toxicity of biodegraded products was assessed on the growth performance of *Zea mays* using seed germination and greenhouse experiment and in vitro cytotoxic effect against normal Vero cells. Data showed the toxicity of biodegraded products was noticeably decreased as compared with that of cypermethrin before degradation.

## 1. Introduction

Chemical insecticides, pesticides, and heavy metals are considered to be among the main pollution sources for soil, groundwater, and other water ecosystems [1,2]. Although these chemical compounds have crucial roles in saving crops and human health by controlling the plant infectious pests and suppressing the vectors that cause different diseases such as malaria and other household insects [3], the overuse of these compounds causes their long-term presence in the soil and directly disrupts the soil microbiota and terrestrial invertebrates, and indirectly has negative impacts on human health through contaminants interfere within the food chain and natural resources [4]. Furthermore, the resistance of insects and pests is increased due to the excessive usage of these chemical compounds. Therefore, the development of new compounds to modifying, destroying, and/or repellence of specific insects or pests are taking more attention from researchers [5,6].

Synthetic pyrethroid compounds are consumed globally as safe insecticides since a few decades ago [7]. Because of low pyrethroid toxicity and its efficacy to protect different crops, farmers commonly use synthetic pyrethroid compounds [8]. Because of low toxicity to mammalian cells, high insecticidal efficacy, low environmental persistence, and relatively low potential to induce resistance of insects, the pyrethroid compounds are represented by 20% of the World’s insecticide markets [9]. The hydrophobic characteristic of pyrethroid compounds not only causes the tight binding with soil organic matter and particles, but also prevents the passage of these compounds to groundwater and thus forming residues that ultimately decrease soil fertility, hinder plant growth, and disturb the soil microbiota [10,11,12].

Cypermethrin (cyano-(3-phenoxy phenyl)methyl]3-(2,2-dichloroethenyl)-2,2-dimethylcyclopropane-1-carboxylate) is a photostable synthetic pyrethroid insecticide used to suppress the growth of pests in vegetables and cotton as well as to control pests indoors and outdoors [13]. Its acts by destroying the central nervous system of insects via producing a hyperexcitable state by interacting with sodium channels [14]. Among pests that are highly affected by cypermethrin are moth pests of fruits, cotton, and vegetable crops [15]. The persistence of cypermethrin in the environment varies from 14.6 to 76.2 days (half-life) dependent on the physicochemical properties of soil [16]. Cypermethrin is highly toxic to aquatic invertebrates and fishes; moreover, it can cause endocrine disruption, neurotoxicity, and immunotoxicity [17]. To decrease the environmental and public health risks related to the usage of pyrethroid, it is necessary to find cost-effective, rapid, and eco-friendly methods to remove or minimize the persistence of insecticides in the environment. Interestingly, cypermethrin can be degraded by soil microbiota such as *Serratia plymuthica*, *Pseudomonas*, *Aspergillus niger*, and *Streptomyces* to reduce environmental injury [13,18].

The biological approach is a promising and innovative strategy used to degrade contaminants based on the catabolic activity of pesticide-degrading microbes [16,19]. Catalytic microbial action is responsible for the degradation, immobilization, eradication, or detoxification of wastes and hazardous materials from the surrounding environments [20,21]. Therefore, bioremediation is a sustainable technology that has been become the most important and dominant way for removing toxic substances from the environment [22,23].

The ester linkage in cypermethrin can be hydrolyzed by *Bacillus* sp. SG2 to form 3-(2, 2-dichloro ethenyl)—2,2-dimethyl-cyclopropanecarboxylate and α-hydroxy-3-phenoxybenzeneacetonitrile. The latter one can be oxidized to 3- phenoxybenzaldehyde which is subsequently converted to 4-hydroxybenzoate. The intermediate compounds were subjected to various oxidation processes to form aliphatic compounds that have low molecular weight [24].

This work aims to study the efficacy of bacterial strain isolated from a contaminated soil sample to degrade pyrethroid insecticide cypermethrin. To achieve this goal, the following steps were achieved: (1) Isolation of bacterial community from collected contaminated soil sample and selection of the most effective bacterial isolate based on high efficacy of cypermethrin degradation. (2) Identification of the most effective bacterial isolate based on it is morphological and physiological characters and confirmed their identification by amplification and sequencing of 16S rRNA. (3) The environmental factors affecting on biodegradation of cypermethrin including incubation period and incubation conditions (static and shaking), incubation temperature, pH values, different cypermethrin concentrations, inocula volume, and different carbon and nitrogen sources were investigated. (4) The efficacy of the most potent bacterial isolate to degrade cypermethrin in the soil was investigated using HPLC and GC–MS analysis. (5) The toxicity of cypermethrin before and after bacterial degradation was assessed by seed germination, a greenhouse experiment to monitor the growth performance of the *Zea mays* plant, and in vitro cytotoxicity on the normal Vero cell line.

## 2. Results and Discussion

### 2.1. Isolation and Screening of Cypermethrin Degrading Bacteria

In the current study, thirteen different bacterial isolates (HIS1–HIS13) were obtained from the contaminated soil samples and showed varied efficacy to grow on mineral salt (MS) agar media supplemented with 500 ppm of cypermethrin. The qualitative screening of these isolates on different cypermethrin concentrations (500 to 4000 ppm as only carbon source) showed that five bacterial isolates of HIS3, HIS4, HIS7, HIS9, and HIS12 displayed the varied growth up to 3500 ppm, while the growth of remaining bacterial isolates was inhibited at a concentration of 1500 ppm. Therefore, the sublethal dose of 2500 ppm was selected to investigate the efficiency of these five isolates to quantitively degrade cypermethrin. Data analysis showed that the maximum degradation percentages were achieved after 8 days. The bacterial isolates exhibit degradation percentages of 47.7 ± 0.04%, 54.3 ± 0.06%, 57.7 ± 0.09%, 27.4 ± 0.05%, and 35.5 ± 0.03% for HIS3, HIS4, HIS7, HIS9, and HIS12, respectively (Figure 1). Based on the obtained data, the bacterial isolate designated as HIS7 was selected as the most effective isolate that exhibited the highest cypermethrin degradation and was used for further investigation.

Consistent with the present study, out of 26 bacterial strains isolated from contaminated wastewater sludge, one bacterial strain exhibits high efficacy to degrade 50 ppm of cypermethrin with percentages of 92.1% after 24 h and 100% after 30 h [13]. The best methods for isolation of pyrethroid-degrading bacteria are achieved from samples collected from contaminated fields with insecticides or other chemicals. For example, high pesticide-degrading *Micrococcus* sp. was isolated from pesticide-contaminated soils [25]. Moreover, pyrethroid-degrading *Sphingobium* sp. was isolated from contaminated wastewater collected from a pyrethroid manufacturing company [26]. In these harsh contaminated fields, the different microbial strains can mineralize, and they utilized the contaminants as their only carbon and energy source.

### 2.2. Identification of the Most Effective Bacterial Isolate HIS7

The morphological characterization of bacterial isolate HIS7 showed that it is a Gram-positive, rod shape, endospore former, and motile. Moreover, the physiological characterization demonstrates the ability of HIS7 to produce catalase enzyme and ferment different sugars, such as glucose, sucrose, and fructose, to produce acids. The bacterial isolate does not exhibit any growth at 10% NaCl. According to morphological and physiological characterizations, the isolate HIS7 belongs to *Lysinibacillus* sp.

The amplification and sequencing analysis of the 16S rRNA gene showed that the bacterial isolate HIS7 has 99% similarity with *Lysinibacillus cresolivorans* (accession number: NR_145635). Based on gene sequence analysis, the phylogenetic tree (Figure 2) showed that the bacterial isolate HIS7 cluster to *Lysinibacillus cresolivorans* group. According to morphological, physiological, and molecular identification, the obtained isolate in the current study has been identified as *Lysinibacillus cresolivorans* strain HIS7. The obtained sequence was deposited in GenBank under the accession number of MZ413355.

*Lysinibacillus* spp. are characterized by their efficacy in bioremediation processes, for example, *Lysinibacillus sphaericus* was selected as the most potent isolate to petroleum hydrocarbon biodegradation and biosurfactant production [27]. The biodegradation of pyrethroid compounds is achieved by different bacterial species such as *Serratia plymuthica, Micrococcus* sp., *Pseudomonas* sp., *Bacillus* sp., *Brevibacterium* sp., *Acinetobacter* sp., and *Acidomonas* sp. [25,28]; fungal species such as *Aspergillus niger, Candida* spp., and *Trichoderma* sp., [28]; and actinomycetes such as *Streptomyces* sp. HU-S-01 [13]. Recently, a bacterial consortium consisting of *Lysinibacillus xylanilyticus*, *Bacillus cereus*, *Lysinibacillus* sp., and *Bacillus* sp. was used for Esfenvalerate insecticide degradation [29]. To the best of our knowledge, this is the first report for biodegradation of Pyrethroid insecticide cypermethrin by *Lysinibacillus cresolivorans*.

### 2.3. Optimizing Factors Affecting the Biodegradation Process

The biotechnological processes depending on biological entities are usually correlated with the environmental conditions which may have positive or negative impacts [30,31]. Therefore, the effects of the environmental parameters on biodegradation of cypermethrin using *Lysinibacillus cresolivuorans* strain HIS7 were investigated. Among these parameters, incubation period at different incubation conditions (shaking and static), incubation temperature, pH values, cypermethrin concentration, bacterial inocula size, and different carbon and nitrogen sources. At each parameter, the biodegradation percentages in the presence and absence of bacterial growth, as well as *Lysinibacillus cresolivuorans* growth was estimated.

The bacterial strain HIS7 was grown in liquid MS media supplemented with 2500 ppm of cypermethrin and incubated at the static and shaking conditions for 10 days. The highest biodegradation percentage was 64.19% after 8 days of incubation at the static condition as compared with those of 53.7% after 7 days at shaking condition (150 rpm) (Figure 3A,B). At the highest degradation percentages, the bacterial growth as measured by optical density (O.D.) reached 1.66 and 1.24 for static and shaking conditions, respectively, as compared to bacterial growth on cypermethrin-free MS, which was 0.68 and 0.25 on static and shaking conditions, respectively (Figure 3A,B). *Bacillus thuringiensis* SG4 showed high efficacy to degrade 78.9% of cypermethrin added to MS media with a concentration of 50 ppm after 15 days of incubation [32].

On the static condition, the cypermethrin acts as a sole oxidant or electron acceptor, which its reduction rate is governed by the formation rate of the electron donor; therefore, the biodegradation rate of cypermethrin has been increased [33]. Compatible with our study, Xiao et al. [34] reported that *Pseudomonas* sp. can biodegrade cypermethrin within 7 days under shaking conditions, whereas the *Bacillus* sp. strain isolated from pesticide-contaminated soil can degrade cypermethrin within 15 days under shaking conditions [35].

The results represented in Figure 3C showed that the biodegradation percentages were increased gradually up to 35 °C and decreased before and after this point. The biodegradation ratio was 62.5% at 35 °C while it reached 6.0% at 45 °C. These results indicated that the optimum growth of *Lysinibacillus cresolivuorans* strain HIS7 at 35 °C (O.D. = 1.42). Furthermore, the activity of some enzyme(s) responsible for cypermethrin breakdown such as hydrolase enzyme was highest at mesophilic temperature [36]. In agreement with our study, the optimum temperature range for biodegradation of pyrethroid insecticide cypermethrin by *Streptomyces* sp. strain HU-S-01 was accomplished at 30–35 °C [13]. High temperature has negative impacts on bacterial cypermethrin degradation due to its destruction of the cell viability and hence suppresses the bacterial enzymatic activity [37]. At MS liquid media containing cypermethrin without HIS7 strain inoculation (control), the maximum abiotic degradation was 3.03% at 35 °C.

The value of pH has a vital role to control the optimal activity of bacterial cultures; moreover, it is the main factor that affects nutrient transport across the bacterial cell membrane [38]. The obtained result showed that the degradation percentage was reached to 60% at pH 7 (Figure 3D). The optimal initial pH value for the breakdown of cypermethrin was 7.0–8.0. Similar results were reported by Cycoń et al. [39] who found that the optimal pH value for enhancing the rapid degradation of cypermethrin by two strains of *Serratia* spp. was achieved at alkaline pH in contrary to low degradation ratio at an acidic pH condition.

However, the efficacy of *Lysinibacillus cresolivuorans* HIS7 to degrade pyrethroid insecticide cypermethrin was concentration-dependent (Figure 4A). At low cypermethrin concentrations ranging from 100 to 1000 ppm, the degradation percentages were in the range of 97.9% to 82.4% after 8 days of incubation. On the other hand, at high concentrations (1500–3500 ppm), the degradation percentages decreased to the ranges of 81.2% and 59.3% (Figure 4A). At sub-lethal dose (2500 ppm), the biodegradation percentage was 70.5%. The time- and dose-dependent biodegradation can be attributed to the microorganisms taking more time to adapt to higher concentrations of pollutants and then the degradation process starts to increase [40]. The efficacy of *Pichia anomala* strain HQ-C-01 to degrade pyrethroid insecticide carbofuran was concentration-dependent; the degradation was increased at low insecticide concentration [41].

Another important factor effect on biodegradation process using bacterial strain is inoculum size. The effective inoculum size of strain HIS7 causing the highest biodegradation of cypermethrin was 3 mL/100 mL of MS liquid media (Figure 4B). The biodegradation efficiency was decreased by increasing or decreasing the inoculum size. The obtained results agree with data recorded by Yin et al. [42] who reported that the best inocula size for biodetoxification of cypermethrin was 3 mL of *Rhodobacter sphaeroides* inoculated into MS liquid media.

The addition of different carbon and nitrogen sources to degradation media has significant effects on the efficacy of bacterial species in biodegradation [43,44]. Data showed that the highest biodegradation percentage (75.9%) was attained after the addition of glucose as an extra carbon source to liquid media (Figure 4C). This could be attributed to the highest bacterial growth (O.D. = 1.5) in presence of glucose. Gurjar and Hamde [45] reported that the biodegradable activity of cypermethrin by *Pseudomonas aeruginosa* was increased in presence of extra carbon sources that enhance the bacterial growth. On the other hand, the addition of glycerol as an extra carbon source led to reducing the degradation percentages to 54.2%. The decreasing of biodegradation percentages in the presence of extra carbon sources with an increase of bacterial growth can be attributed to catabolite repression caused by extra carbon sources [46,47]. Interestingly, the *L. cresolivuorans* strain HIS7 exhibits high efficacy to degrade cypermethrin in the presence of different nitrogen sources. Data showed that the optimal nitrogen source for biodegradation of cypermethrin using strain HIS7 was NH_4_Cl_2_ with degradation percentages of 69.2% as compared with the liquid media contains cypermethrin only (29.3%) (Figure 4D). Previous studies reported that the addition of extra nitrogen sources harmed insecticide degradation by bacteria in the soil [48,49].

### 2.4. Comparison between Biodegradation Percentages before and after the Optimization Process

The optimization of different environmental factors has positive impacts on biodegradation processes as reported in different published studies [44,45]. The biodegradation percentages due to the bacterial growth in MS liquid media containing 2500 ppm of cypermethrin under all previously optimized conditions were assessed. A significant increase was recorded in the biodegradation percentages (86.9%) after optimization of environmental factors as compared with those (57.7%) before optimization (Figure 5). Moreover, the bacterial growth measured by O.D. was increased due to optimization from 1.4 to 1.85. The obtained data were compatible with those of Gurjar and Hamde [45], who found that the biodegradation percentages of cypermethrin increased after optimization of environmental factors compared with those before optimization.

### 2.5. Biodegradation Assay of Cypermethrin in Soil Using HPLC

The bacterial efficacy of cypermethrin degradation in soil was investigated either by a qualitative method using spectrophotometrically at interval times (7, 14, 21, 28, 35, and 42 days) or by a quantitative method using HPLC after 42 days. Qualitative analysis exhibits the efficacy of *L. cresolivuorans* strain HIS7 to degrade cypermethrin in soil was time-dependent (Figure 6A). The degradation percentages of cypermethrin in soil using *L. cresolivuorans* strain HIS7 was increased from 54.7% after 7 days to 93.1% after 42 days as compared with control (cypermethrin in soil without bacterial inoculation), which was 3.9% after 7 days and reached 8.2% after 42 days (Figure 6A). The HPLC analysis of the obtained extract from control soil amended with cypermethrin at a concentration of 2500 mg kg^−1^ showed one peak appeared at retention time (R.T.) of 3.460 min. While the HPLC chart of the extract obtained from treated sterilized soils and inoculated by *L. cresolivuorans* HIS7 displayed three peaks at R.T. of 2.510, 2.878, and 3.230 min. The analysis showed that the width, height, and area of the single peak obtained from the control sample were 0.043, 356.634, 3486.857, respectively, whereas the three peaks that appeared in HPLC spectra due to the bacterial treatment showed the width of (0.578, 0.133, and 0.782), the height of (29.231, 3.015, and 0.391), and peaks area of (450.504, 13.905, and 89.401), respectively.

### 2.6. GC-MS Analysis of Cypermethrin Biodegradation

Cypermethrin (2500 mg kg^−1^) from the inoculated soil with *L. cresolivuorans* strain HIS7 was extracted and underwent GC–MS analysis to detect the biodegradable products as compared with control (soil sample containing cypermethrin without bacterial incubation and incubated for the same time). The extraction was accomplished for soil samples incubated for 42 days. The GC–MS chart for control exhibits the main peak of cypermethrin residue at R.T 17.08 min with a percent area of 31.51% (Table 1; Figure 7A). The major peaks of different compounds resulting from cypermethrin biodegradation by *L. cresolivuorans* HIS7 appeared at varied R.T. of 4.43, 9.20, 10.92, 11.39, and 11.80 min (Table 1; Figure 7B). The main biodegradable products are defined as acetic acid (4-chloro-2-methylphenoxy); 1H-purine-2,6-dione,3,7-dihydro-1,3,7trimethy; 9-octadecenamide; benzene ethanamine, à-methyl-3-[4-methylphenyloxy]; and 1,2-Benzenedicarboxylicacid-3-nitro with percent peak area of 0.20%, 4.37%, 1.0%, 0.9%, and 1.87%, respectively (Table 1). The utilization of GC–MS analysis to estimate the biodegradable products due to the breakdown of cypermethrin and other contaminants was achieved by various investigators [35,50,51]. The biodegradable compound of 1,2-benzenedicarboxylicacid was detected from GC-MS analysis of biodegradation of permethrin by *Acinetobacter baumannii* strain ZH-14 [52]. Based on GC–MS analysis, the cypermethrin can be metabolized by bacterial strain *L. cresolivuorans* HIS7 to different intermediate compounds with the potential to low toxicity.

### 2.7. Toxicity Assessment

#### 2.7.1. Seed Germination of *Zea mays*

Seed germination assay is a sensitive, most common, rapid, and an effective method to evaluate the toxicity of hazardous compounds [53,54]. Therefore, the toxicity of cypermethrin and its biodegradable products were assessed through seed germination of *Zea mays*. Data analysis showed that cypermethrin (negative control) had a highly toxic effect on the germination process as compared to those after bacterial degradation. The root length of *Zea mays* treated by cypermethrin without bacterial inoculation was 11.0 ± 0.06 cm, whereas the length of treated root by pyrethroid insecticide cypermethrin after degradation by *L. cresolivuorans* HIS7 was 25.0 ± 0.1 cm similar to those treated by distilled H_2_O (29.0 ± 0.1 cm) (Table 2; Figure 8). The root length of *Zea mays* treated by biodegradable products and distilled H_2_O was increased with percentages of 56% and 62.1%, respectively, than those treated by cypermethrin (negative control). 

On the other hand, the fresh and dry weights of *Zea mays* roots treated by distilled H_2_O, cypermethrin without degradation, and biodegradable products were (1715.4, 862.5, and 1655.5 mg) and (505.2, 116.1, and 410.4 mg), respectively (Table 2). Similarly, different types of pesticides exhibit negative impacts with a varying degree on the seed germination of various plants such as chili, *Brassica nigra, Solanum melongena*, tomato, maize, Cowpea, and *Typha latofolia* [55,56,57]. Seed germination of *Vigna unguiculata* treated with cypermethrin is increased at low concentration and decreased with higher concentration [57]. This phenomenon could be attributed to the highly toxic effects of pyrethroid insecticides on meristematic cells at high concentrations [57]. In this regard, the chlorsulfuron insecticide exhibit toxic effects on the root cellular structure of *Phaseolus vulgaris*, *Pisum sativum*, and *Vicia faba* through the delay the division of radical cells and hence suppress the growth of roots [58]. Furthermore, the seed germination of the tomato plant was decreased due to treatment with a high concentration of pesticides of emamectin benzoate, lambda-cyhalothrin, imidacloprid, and alpha-cypermethrin [56]. The seed germination of tomato was completely inhibited at the concentration of 160, 240, 2000, and 500 ppm from emamectin benzoate, lambda-cyhalothrin, imidacloprid, and alpha-cypermethrin, respectively after 27 h. Although seed germination was recovered with time passage, it is still lower than control [56].

#### 2.7.2. Greenhouse Experiment

Besides seed germination, the toxicity of cypermethrin and its biodegradable products were also investigated through the growth performance of *Zea mays* plant in the soil (Figure 8; Table 3). Data analysis showed that the root length of *Zea mays* treated with distilled water and biodegradable products were 14.0 ± 0.6 and 12.0 ± 0.6 cm, respectively which significantly compared with those treated with cypermethrin without bacterial degradation (5.0 ± 0.4 cm). On the other hand, there is no significant difference between shoot length of positive control (122.0 ± 0.7 cm) and those treated by biodegradable products (115.0 ± 0.7 cm), whereas it is significantly (*p* ≤ 0.05) different as compared with shoot length of negative control (88.0 ± 0.5 cm) (Table 3). 

The growth performances of *Zea mays* were varied according to different treatments. Data analysis showed that the fresh and dry weights of roots treated by distilled H_2_O, cypermethrin without degradation, and biodegradable products were recorded (3.6 ± 1.5, 2.1 ± 1.0, and 3.1 ± 1.1 g) and (1.9 ± 0.4, 1.1 ± 0.6, and 1.4 ± 0.2 g), respectively, whereas the fresh and dry weights of the shoot system were as follows: (21.5 ± 2.3 and 12.9 ± 2.2), (13.2 ± 1.1 and 5.3 ± 0.9), and (19.9 ± 2.3 and 11.9 ± 3.1) for plants irrigated with distilled H_2_O, cypermethrin without degradation, and biodegradable products, respectively (Table 3). It is suggested that the toxicity of cypermethrin after biodegradation using bacterial strain HIS7 was decreased as compared with their toxicity before bacterial degradation.

The biodegradation efficiency of cypermethrin in soil depending on different factors such as degradation activity of soil microorganisms, soil texture, moisture content, temperature, pH, and organic matter present in the soil [59]. Moreover, the biological habitats of agricultural soils, such as qualitative and quantitative changes in microbiota, activity of nitrogen-fixing microorganisms, and all over the distribution of rhizospheric microbial community, are detrimentally affected due to the overusing of pyrethroid insecticides. All these factors have harmful effects on soil fertility and hence plant growth performance [60,61]. In the same regard, the morphological characteristics (shoot and root length) as well as the photosynthetic pigments (chlorophyll a, b, and carotenoid) of *Allium cepa* L., *Zea mays* L., and *Lathyrus sativus* L. were highly effected due to the treatment with different concentrations (0.2, 0.4, 0.6, and 0.8 g L^−1^) of cypermethrin. Data showed that the shoot and root length was significantly decreased (*p <* 0.05) as compared with non-treated plant. Moreover, the concentration of photosynthetic pigments was decreased by increasing the cypermethrin concentration [62].

#### 2.7.3. In Vitro Cytotoxic Efficacy

Finally, the toxicity of cypermethrin before and after bacterial degradation was assessed on normal Vero cell line. The in vitro cytotoxic efficacy was evaluated by the MTT assay method. In the same regard, Suman and co-authors used the MTT assay method to evaluate the in vitro cytogenetic efficacy of cypermethrin on human lymphocytes [63]. Furthermore, the cytotoxicity and genotoxicity efficacy of Furia^®^ 180 SC (zeta-Cypermethrin) and Bulldock 125^®^ SC (β-cyfluthrin) pyrethroid insecticides on human peripheral blood lymphocytes was investigated using the MTT assay method [64]. This method is a highly sensitive colorimetric used assay to evaluate the cell viability and cellular toxicity due to exposure to external substances [65,66,67].

The results showed that the toxicity of cypermethrin before and after bacterial degradation was dose-dependent. Consistent with our result, Kakko et al. [68] reported that the toxicity of cypermethrin on SH-SY5Y was dose-dependent. The viability of the Vero cell line due to the treatment with 2500 ppm of cypermethrin without bacterial degradation was 2.6%, while the viability reached 79.8% after being treated with biodegradable products at the same concentration (Figure 9). By decreasing the concentration, the viability increased till reached 5.9% and 97.9% due to treatment with 625 ppm of cypermethrin before and after bacterial degradation, respectively. The cell viability reached 100% after treatment with 159 ppm of biodegradable products and 9 ppm of cypermethrin without bacterial degradation (Figure 9). The obtained data revealed that the toxicity of biodegradable end products due to biodegradation of pyrethroid insecticide cypermethrin by *L. cresolivuorans* strain HIS7 is highly decreased as compared with cypermethrin without degradation. Cypermethrin exhibits cytotoxic efficacy against the HepG2 cell line, and the cell viability was decreased based on concentrations of cypermethrin used [69]. This cytotoxic efficiency could be attributed to apoptosis or necrosis caused due to cypermethrin exposure [70]. Moreover, the cypermethrin showed a reduction in the viability of CHO-K1 and DET551 (human fibroblast) cell line [71,72], and attributed this reduction to redox deficiency which ultimate to oxidative stress and production of reactive oxygen species (ROS) that damage various biomolecules such as lipid, nucleic acids, and proteins [73]. The different cytotoxic efficacy of cypermethrin on varied cell lines may be attributed to the degree of purity [74].

## 3. Materials and Methods

### 3.1. Reagents and Materials Used

Cypermethrin (95% purity) was obtained from Ali Akbar group, 1-KM Bhoptian Chowk Defence Road, Off Raiwind Road, Lahore, Pakistan. A stock solution of cypermethrin (10,000 ppm) was prepared by dissolving 1052 mL of cypermethrin in a 100 mL methanol solution. Methanol, crystal violet, and other chemical reagents used in this study were analytical grades obtained from Sigma Aldrich, Cairo, Egypt. Mineral Salt (MS) media containing (g L^−1^) 1.0, NH_4_NO_3_; 1.0, NaCl; 1.5, K_2_HPO_4_; 0.5, KH_2_PO_4_; 0.2, MgSO_4_·7H_2_O, pH 7.0. Agar was supplemented as 1.5% (*w*/*v*) for medial solidification. The sterilization of media was done at 121 °C for 20 min before using.

### 3.2. Isolation of Cypermethrin 95% Degrading Bacteria

The cultivated soil sample collected from El-Menofia Governorate (GPS: 30°30′22.45″ N; 31°0′15.3″ E) was used for bacterial isolation. The collected soil sample has a history of chemicals spray including herbicides, insecticides, and nematicides. The isolation procedure was achieved using MS agar media supplemented with 500 ppm of cypermethrin as a sole carbon and energy source. Under aseptic condition, one milliliter of diluted sample (10^3^) was spread over the surface of Petri dishes of the prepared MS agar media and incubated at 35 °C for seven days.

The different bacterial colonies that appeared on the Petri plates were picked up and reinoculated again onto a new MS agar plate to obtained purified isolates which were preserved onto MS agar slants and preserved at −4 °C for further work [75]. 

### 3.3. Qualitative Screening for Bacterial Isolates

The efficacy of obtained bacterial isolates to degrade cypermethrin were screened at 35 ± 2 °C using MS agar media supplemented with different concentrations (500 to 4000 ppm) of cypermethrin. The presence and absence of bacterial growth was recorded as + and − after incubation period of 8 days. 

### 3.4. Quantitative Screening

The most potent bacterial isolates (based on the highest tolerate of cypermethrin concentrations) were screened to investigate their efficacy to degrade high cypermethrin concentration spectrophotometrically (Shimadzu UV-2410, Kyoto, Japan) as follows: the bacteria isolate was inoculated at 35 ± 2 °C into MS broth media supplemented with 2500 ppm (as a sub-lethal dose) of cypermethrin for 10 days. The samples were analyzed every 2 days to extract and assess the biodegradation ratio as follows. 

#### 3.4.1. Extraction of Cypermethrin Residues

Cypermethrin residue assay in the inoculated MS broth media was achieved according to Zhang et al. [23] with slight modification. Cypermethrin concentration was assessed for the withdrawn samples at 2 days-time intervals. The cypermethrin residue was extracted from the medial broth with equal volume (1:1 *v*/*v*) of hexane. The extract was collected and mix with dried anhydrous sodium sulfate, then the previous mixtures are exposed to nitrogen gas for dryness and intense concentration using rotary evaporator at room temperature. The resultant dissolved in 5 mL of hexane for further investigation.

#### 3.4.2. Degrading Assay Method

According to Lin et al. [13], a spectrophotometric method was used for the evaluation of trace levels of cypermethrin insecticide. In this method, the cypermethrin was hydrolyzed to cyanide ion; these ions react with potassium iodide and leuco crystal violet (LCV) producing a crystal violet dye. The LCV was prepared by mixing 200 mL distilled water with 3 mL of 85% phosphoric acid and 250 mg of leuco-crystal, (4,4′,4″ methylidynetris (N,N’dimethylaniline) (CH[C_6_H_4_N(CH_3_)_2_]_3_). Dye of the previous mixture was dissolved by shaking, and the flask content was diluted to 1 L. An aliquot of test solution containing cypermethrin 95% was added to a 25 mL graduated cylinder and mixed with 1.0 mL of 20% sodium hydroxide. The mixture complete hydrolysis was achieved at room temperature after 10 min. One milliliter of 0.1% potassium iodide was put in an acidic medium to release iodine; afterwards, 1.0 mL of the prepared LCV was supplemented with continuous shaking for 15 min until the formation of complete color. The absorbance of produced color was assessed at 595 nm. The degradation percentages were calculated using the following equation [51]:(1)$$\mathrm{Degradation}\;\mathrm{percentages}\;(\%)=\frac{C_{i}-C_{f}}{C_{i}}\times 100$$
where C_i_ refers to the initial absorbance of cypermethrin and C_f_ refers to final absorbance of cypermethrin. 

### 3.5. Identification of Cypermethrin Degrading Bacteria

The most potent cypermethrin degrading bacteria were designated as HIS7 which showed the highest biodegradation efficacy of cypermethrin. Morphological and biochemical tests were used to identify bacterial isolate of HIS7. Molecular identification was achieved based on PCR-sequencing analysis of the 16S rDNA gene. Genomic DNA was extracted, and 16S rDNA gene fragment was amplified using primers set of 27f (5-AGAGTTTGATCCTGGCTCAG-3) and 1492r (5-GGTTACCTTGTTACGACTT-3). The PCR amplification was conducted on 5 ng of bacterial genomic DNA using 1 × PCR buffer, 0.5 mM MgCl_2_, 2.5 U Taq DNA polymerase (QIAGEN Inc., Germantown, MD, USA), 0.25 mM dNTP (Deoxynucleoside triphosphate), and 0.5 µM primer [76].

PCR amplification was conducted at 94 °C for 3 min, followed by 30 cycles at 94 °C for 0.5 min, 55 °C for 0.5 min, 72 °C for 1 min, and a final extension exhibited at 72 °C for 10 min. The 16S rDNA sequence was performed by Beijing Liuhe Huada Genomics Company (Beijing, China). Multialignment of the 16S rDNA sequence and the highly similar sequences was performed by CLUSTAL W. Phylogenetic analysis was conducted based on the kimura 2-parameter and the tree was built by the neighbor-joining method using MEGA 4.0 software with bootstrap value of 1000 replicates [77,78].

### 3.6. Effects of Environmental Factors on Biodegradation of Cypermethrin 95%

The optimal environmental conditions for biodegrading of cypermethrin 95% by HIS7 strain was examined using single-factor test. In each factor, the bacterial growth was designated as optical density (O.D.) and the degradation ratio was calculated. Different concentrations of cypermethrin 95% ranging from 100 mg/L to 3500 mg/L have been studied. To examine the effect of temperature and pH on biodegrading of cypermethrin by HIS7, the different temperature degrees of (20, 25, 30, 35, 40, and 45 °C) and pH values of 3.0–11.0 were used.

Different types of nitrogen sources represented by (NaNO_3_, NH_4_Cl_2_, urea, peptone, Aspartic acid, and Glutamic acid) and different carbon sources of glucose, maltose, glycerol, starch, and lactose have been studied to investigate their efficacy of biodegradation. Non-inoculated media were conducted at the same conditions and the experiment units were performed in triplicates.

### 3.7. Biodegradation of Cypermethrin in Soil

#### 3.7.1. Inocula Preparation

The bacterial strain HIS7 was cultured in a 250 mL conical flask with 120 mL of nutrient broth media and incubated at 35 ± 2 °C. At the exponential phase, the cell pellets were collected by centrifugation (5000× *g*, 10 min), washed 3 times with 50 mL of KH_2_PO_4_-K_2_HPO_4_ (0.15 mol/L, pH 7.0). The washed pellets were resuspended at phosphate buffer and used as a soil inoculant. The experimental treatments were constructed along with non-inoculated control samples in triplicates to avoid the effects of hydrolysis and photolysis. 

#### 3.7.2. Soil Samples

The soil samples were collected from Abo-Rawash, Giza governorate, Egypt (GPS: 31°04′30″ N; 55°01′30″ E). The soil sample (g kg^−1^ dry weight) contained organic matter 32, total nitrogen 1.38, total phosphorus 1.02, and a pH value of 6.4. Samples were air-dried at room temperature, mixed thoroughly, and sieved to 2 mm in size, then stored in sealed polyethylene bags at 4 °C for further use [79].

#### 3.7.3. Inoculation and Extraction Method

Approximately, 100 g of sterilized soil was placed in a container, mixed with cypermethrin at a concentration of 2500 mg kg^−1^, inoculated with 50 mL of bacterial suspension, and incubated at 35 ± 2 °C for 42 days. The water content in the soil sample was maintained at 40% of the water holding capacity. A soil sample was withdrawn at different times intervals (0, 7, 14, 21, 28, 35, and 42 days) to detect the cypermethrin residue and its biodegradable products as follows: 5.0 g of inoculated soil was weighed and mixed with 15 mL of acidified acetone (acetone: water: conc. H_3_PO_4_ (98:1:1)). The suspension was stirred and shaken for 4 h, then centrifuged at 4000 rpm for 30 min. The previous steps were repeated three times to obtain all cypermethrin residue in the extract. The different extracts were combined and evaporated using a rotary evaporator at 45 °C to obtain the residue used for HPLC and GC–MS analyses [80].

##### HPLC Analysis of Cypermethrin in Soil

The biodegradation percentages of cypermethrin were analyzed by an SYKAM HPLC achieved with an HPLC Pump S1122, programmable variable-wavelength UV detector S3210, Autosampler S5200, and Phenomenex C18 reversed-phase column (150 mm) in the Regional Center of Mycology and Biotechnology, Al-Azhar University, Cairo, Egypt (Instrument code, LC1620A; Liquid Chromatography, made in Taiwan, China).

For HPLC analysis, the resulted residue was dissolved in 1.0 mL of methanol–water (90:10 *v/v*) for analysis. The detector output was performed by the Clarity chromatography data system. The concentration analysis was processed by gradient elution conditions with initial solvent conditions of 100% solvent A: B (solvent A consisting of methanol: H_2_O: acetic acid with percentages 20:80:0.5 (*v*/*v*), while solvent B consisted of methanol: H_2_O: acetic acid with percentages of 80:20:0.5 (*v*/*v*)) for 10–20 min at a flow rate of 1.0 mL min^−1^. The detection was performed at 230 nm. The sample injection volume was 20 µL. The calibration curves of cypermethrin were made from the serial dilutions of the samples dissolved in 100% methanol. The linear range and the equation of linear regression were obtained sequentially at 0–200 µL intervals at 25 s. The calibration equation was constructed by plotting the mean areas resulted from the standard solution and concentrations. The concentrations of cypermethrin 95% and its degradations ratios were determined based on the peak areas in the chromatograms [81]. 

##### GC/MS Analysis of Cypermethrin and Its Biodegradable Products

Purification of the residues was performed using hexane pre-poured Florisil^®^ columns (Agilent SAMPLIQ Florisil^®^, Santa Clara, CA, USA) and 0.22 m membranes (Millipore, Merck KGaA, Darmstadt, Germany). The GC/MS system (Agilent 7890A/5975, Agilent Technologies, Santa Clara, CA, USA) and electron ionization (EI). EI (70 eV) was adjusted at a trap current of 100 mA and a source temperature of 200 °C to identify the resulted metabolites. Full scan spectra were acquired at *m*/*z* 45–500 at 2 s per scan. Agilent MSD ChemStation software with the Agilent chemical library (Regional Center of Mycology and Biotechnology, Al-Azhar University) was used to confirm metabolites identification using standard MS and data collection.

### 3.8. Toxicity Assessment

#### 3.8.1. Seed Germination

The toxic effect of cypermethrin and its biodegradable products on seed germination of *Zea mays* L. was assessed. The plant seeds were obtained from Agricultural Research Center, Giza, Egypt. The healthy seeds (uniform in size and color) were chosen and undergo surface sterilization as follows: first, washed with distilled water, followed by sterilization with 1% (*v*/*v*) of sodium hypochlorite for approximately 2 min, washed fully again with distilled water, and kept to dry at room temperature [82].

Different treatments were conducted as follows: (1) distilled water which serve as a positive control. (2) Extract of MS broth media supplemented with cypermethrin (2500 ppm) and incubated under optimum condition without bacterial inoculation. The extract was concentrated by a rotary evaporator and dissolved approximately 100 µg of the obtained residue in 100 mL of distilled water. This treatment serves as a negative control. (3) Biodegradable products (extract of MS broth media supplemented with cypermethrin (2500 ppm) inoculated by bacterial strain HIS7 and incubated under optimum condition. The obtained extract was concentrated using a rotary evaporator and dissolved approximately 100 µg of the obtained residue in 100 mL of distilled water). The selected *Zea mays* seeds were germinated before the experiment in distilled water to confirm the health of the seed through the growth of the radical till reach 0.5 cm. Thereafter, three groups (each group containing seven uniforms germinated *Zea mays* seeds) were assembled and set into a petri dish containing Whatman filter paper and incubated at 30 °C. The seeds are irrigated as needed by the previous treatment. Growth parameters (root length and fresh and dry root biomass) were recorded after 12 days [83]. The experiment was achieved in triplicates. 

#### 3.8.2. Greenhouse Experiment

The toxic efficacy of cypermethrin before and after bacterial degradation was monitored on the growth performance of *Zea mays* L (Cultivar Giza 9) using a completely randomized pot experiment. The greenhouse experiment was achieved using sandy soil, their textural was sand (95.6%), silt (2.25%), and clay (1.15%), and their chemical composition was P (22.9 mg kg^−1^), K (16.0 mg kg^−1^), Na (185.6 mg kg^−1^), Ca (26.6 mg kg^−1^), and Cl (128.6 mg kg^−1^). The following treatments were conducted, positive control, negative control, and biodegradable products (see seed germination section). Before the experiment, the *Zea mays* seeds were surface sterilized through soaking into sodium hypochlorite (2.5%) for 3 min and washed with distilled water three times. After that, the sterilized seeds were left to pre-germination to select 3 groups of similar sterilized germination seeds. The experiment was achieved with 5 pots for each treatment and each pot contained three germination seeds. The planting pots were irrigated with the previous treatment as needed and incubated at 25–30 °C for 45 days. At the end of the experiment, the plant was harvested, the shoot and root system were separated, and any soil particles attached to the root system were removed. The plant height, root length, and fresh weight, and dry weight of shoot and root system were calculated [84]. 

#### 3.8.3. In Vitro Cytotoxicity Using the MTT Assay Method

The in vitro cytotoxic efficacy of cypermethrin before and after bacterial degradation was investigated against normal Vero cell line (normal kidney cell from African green monkey) using MTT [3-(4,5-dimethylthiazol-2-yl)-2, 5-diphenyl tetrazolium bromide] viability assay method. The conducted treatments were negative control (cypermethrin at 2500 ppm without bacterial degradation) and biodegradable products (see seed germination section). The cells are obtained from ATCC (American type culture collection). Cells at concentration 1 × 10^5^ cell/mL are inoculated into a 96-well plate containing 0.2 mL media/well and incubated for 48 h at 37 °C after being treated with double-fold concentration (2500–9.7 µg mL^−1^) of cypermethrin before and after degradation. At the end of the incubation period, approximately 30 µL of MTT (5mg/mL dissolved in phosphate buffer) was mixed with each well and incubate the plate at 37 °C, 5% CO_2_ for 5 h. The purple color is formed after adding 1 mL of DMSO due to formazan crystal formation [85]. The experiment was performed in triplicate. The color intensity was measured at 560 nm using an ELISA plate reader and the cell viability was calculated according to the following equation [86].
(2)$$\mathrm{Cell}\;\mathrm{viability}\;(\%)=\frac{\mathrm{Absorbance}\;\mathrm{of}\;\mathrm{the}\;\mathrm{treatment}}{\mathrm{Absorbance}\;\mathrm{of}\;\mathrm{control}}\times 100$$

### 3.9. Statistical Analysis 

One way analysis of variance (ANOVA) was performed on the obtained data to test the significant effect of different treatments using SPSS v17 software (IBM, Armonk, NY, USA). Multi-comparison of Tukey’s test at significant level of *p* ≤ 0.5 was conducted when ANOVA test is significant. 

## 4. Conclusions

In this study, the efficacy of different bacterial species isolated from a contaminated soil sample to degrade the pyrethroid insecticide cypermethrin was investigated. Data showed that the most potent bacterial isolate designated as HIS7 showed high efficiency in degrading cypermethrin insecticide and was identified as *Lysinibacillus cresolivuorans* based on a morphological and physiological test as well as amplification and sequencing of 16S rRNA. Data analysis revealed that the degradation percentages were increased from 57.7% to 86.9% after optimizing the environmental factors of incubation condition (static), incubation period (8-days), incubation temperature (35 °C), pH (7), inocula size (3%), in addition of extra-carbon source (glucose) and nitrogen source (NH_4_Cl_2_), and 2500 ppm of cypermethrin. Moreover, the bacterial strain HIS7 showed a high potential to degrade cypermethrin in the soil as detected by qualitative methods (spectrophotometric method) and quantitative methods (HPLC and GC–MS analyses). Data showed that the degradation percentages of soil extracted sample increased from 54.7% after 7 days to 93.1% after 42 days as compared with control (cypermethrin in soil without bacterial inoculation) which was 3.9% after 7 days and reached 8.2% after 42 days. On the other hand, the HPLC analysis showed one peak at a retention time of 3.460 min for control, while it showed three peaks at R.T. of 2.510, 2.878, and 3.230 min after degradation by *L. cresolivuorans* HIS7. Moreover, the GC–MS for treated soil extract showed five biodegradable products defined as acetic acid (4-chloro-2-methylphenoxy); 1H-purine-2,6-dione,3,7-dihydro-1,3,7trimethy; 9-octadecenamide; benzene ethanamine, à-methyl-3-[4-methylphenyloxy]; and 1,2-benzenedicarboxylicacid-3-nitro with percent peak area of 0.20%, 4.37%, 1.0%, 0.9%, and 1.87% respectively. The toxicity of cypermethrin and biodegradable products was assessed by seed germination of *Zea mays*, growth performance of *Zea mays* under greenhouse experiment and in vitro cytotoxic efficacy on Vero-normal cells. Data showed that the toxicity of biodegradable products was highly decreased as compared with cypermethrin without degradation. The root length of *Zea mays* treated by cypermethrin without degradation was 11.0 ± 0.06 cm, whereas root length treated by biodegradable products 25.0 ± 0.1 cm which is not significant with those treated by distilled H_2_O (29.0 ± 0.1 cm). Furthermore, the growth performance of *Zea mays* plant represented by fresh weight and dry weight of shoot and root exhibit no significant difference between biodegradable products and distilled H_2_O, whereas its high significance with those treated by cypermethrin without degradation. The viability of the Vero cell line due to treatment with 2500 ppm of cypermethrin without bacterial degradation was 2.6%, while the viability reached 79.8% after being treated with biodegradable products at the same concentration. Based on obtained data, the high efficacy of bacterial strain *L. cresolivuorans* HIS7 to degrade pyrethroid insecticide cypermethrin is evident. This study gives more information about the efficacy of biodegradation process as a promising tool to decompose the hazardous materials that persistence in the environment. This process is characterized by cheap, eco-friendly, highly effective, and biocompatible.

## Figures and Tables

**Figure 1 plants-10-01903-f001:**
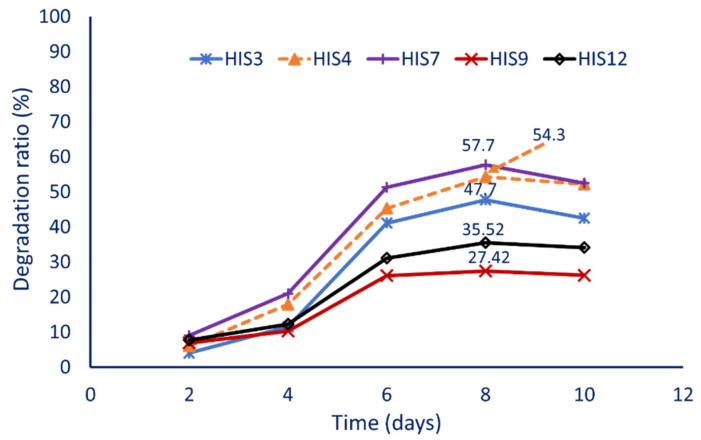
The quantitative screening of five bacterial isolates to degrade the cypermethrin at a concentration of 2500 ppm at different interval times.

**Figure 2 plants-10-01903-f002:**
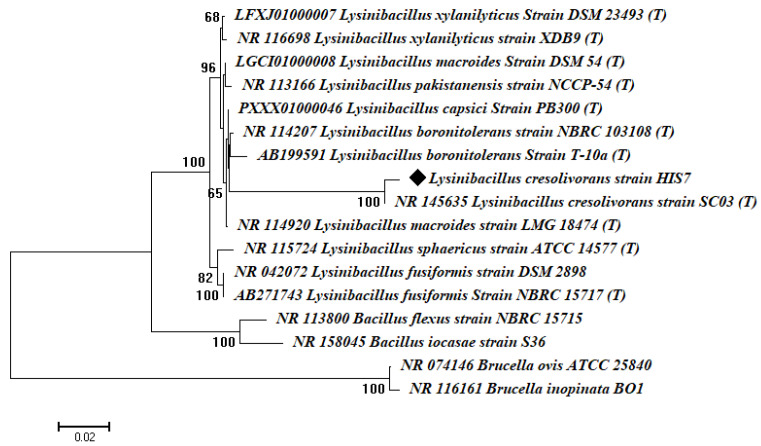
Phylogenetic tree of 16S rRNA sequences of the bacterial isolate HIS7 compared with the retrieved sequences of NCBI GenBank. The matrices of tree reconstruction were determined using a neighbor-joining approach. A bootstrap value greater than 60% (1000 replicates) was indicated on the branch. The number of substitutions per sequence is shown on the scale.

**Figure 3 plants-10-01903-f003:**
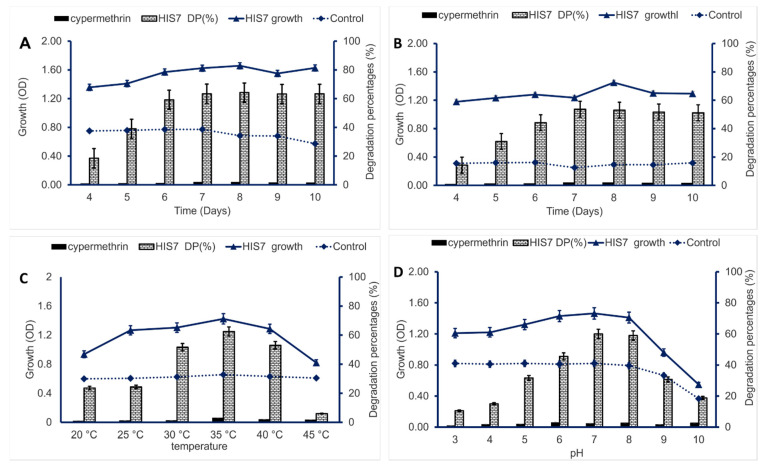
The optimization process of cypermethrin bacterial degradation. Panels (**A**,**B**) denote the optimum incubation periods at static and shaking conditions, respectively; panel (**C**), optimum temperature; panel (**D**), optimum pH values. Cypermethrin meaning the abiotic degradation percentages of insecticide in MS liquid media in absence of bacterial inoculation; HIS7 DP (%) denotes the degradation percentages of cypermethrin in MS liquid media in presence of *Lysinibacillus cresolivuorans* strain HIS7; HIS7 growth denotes the bacteria growth (O.D.) in MS liquid media containing cypermethrin; Control denotes the bacterial growth (O.D.) in MS liquid media in absence of cypermethrin. Data represented by Mean ± SE (n = 3). The standard deviation is less than the size of symbols if no error bars are seen.

**Figure 4 plants-10-01903-f004:**
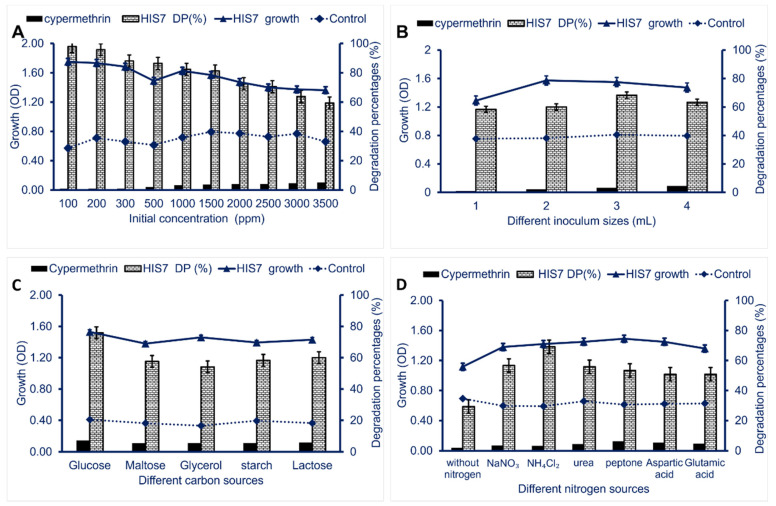
The optimization process of cypermethrin bacterial degradation. Panel (**A**) denotes different cypermethrin concentrations; panel (**B**) denotes the bacterial inoculum size; panel (**C**) denotes different carbon sources, and panel (**D**) denotes different nitrogen sources. Cypermethrin meaning the abiotic degradation percentages of insecticide in MS liquid media in absence of bacterial inoculation; HIS7 DP (%) denotes the degradation percentages of cypermethrin in MS liquid media in presence of *Lysinibacillus cresolivuorans* strain HIS7; HIS7 growth denotes the bacteria growth (O.D.) in MS liquid media containing cypermethrin; Control denotes the bacterial growth (O.D.) in MS liquid media in absence of cypermethrin. Data represented by Mean ± SE (n = 3). The standard deviation is less than the size of symbols if no error bars are seen.

**Figure 5 plants-10-01903-f005:**
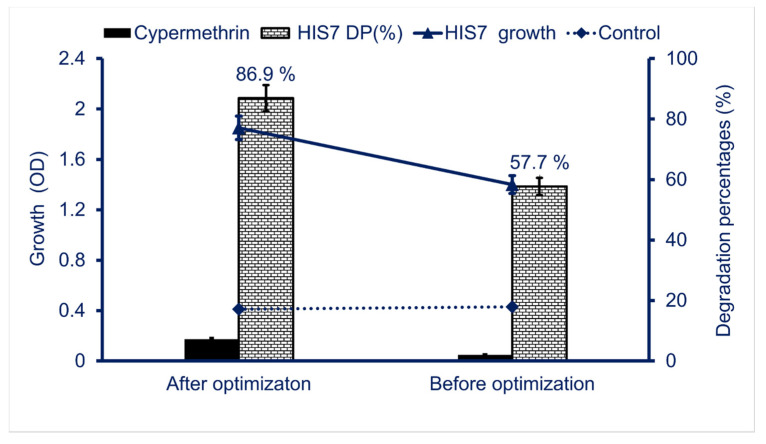
The comparative study between biodegradation percentages of cypermethrin using *Lysinibacillus cresolivuorans* strain HIS7 before and after environmental factors optimizations. The environmental factors before optimization were incubation period 8 days at shaking condition, 2500 ppm of cypermethrin as only carbon source, incubation at 37 °C, and pH value of 7, whereas the environmental factors after optimization were incubation period 8 days at static condition, 2500 ppm of cypermethrin, incubation temperature was 35 °C, pH value was 7, 3% inocula volume, and in presence of glucose and NH_4_Cl_2_ as extra-carbon and nitrogen source, respectively. Data represented by Mean ± SE (n = 3).

**Figure 6 plants-10-01903-f006:**
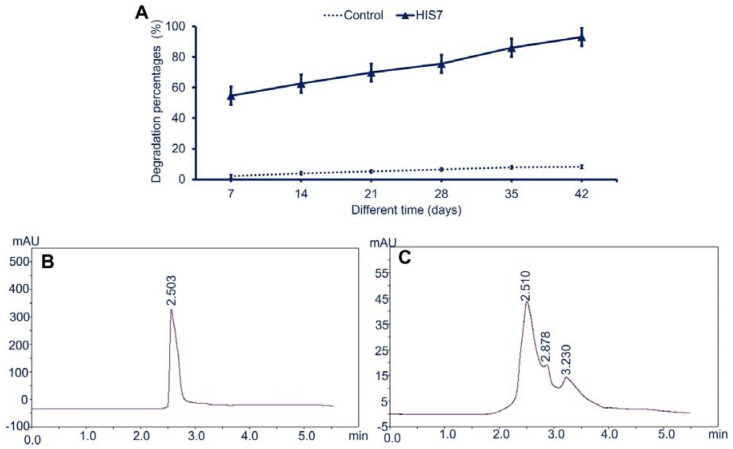
The biodegradation assay of cypermethrin in soil inoculated by bacterial strain HIS7. Panel (**A**) is a qualitative assay of cypermethrin in soil over time using the spectrophotometric method; panels (**B**,**C**) are the HPLC chart of soil cypermethrin in the absence and presence of bacterial inoculation, respectively. Data in panel (**A**) represented by Mean ± SE (n = 3).

**Figure 7 plants-10-01903-f007:**
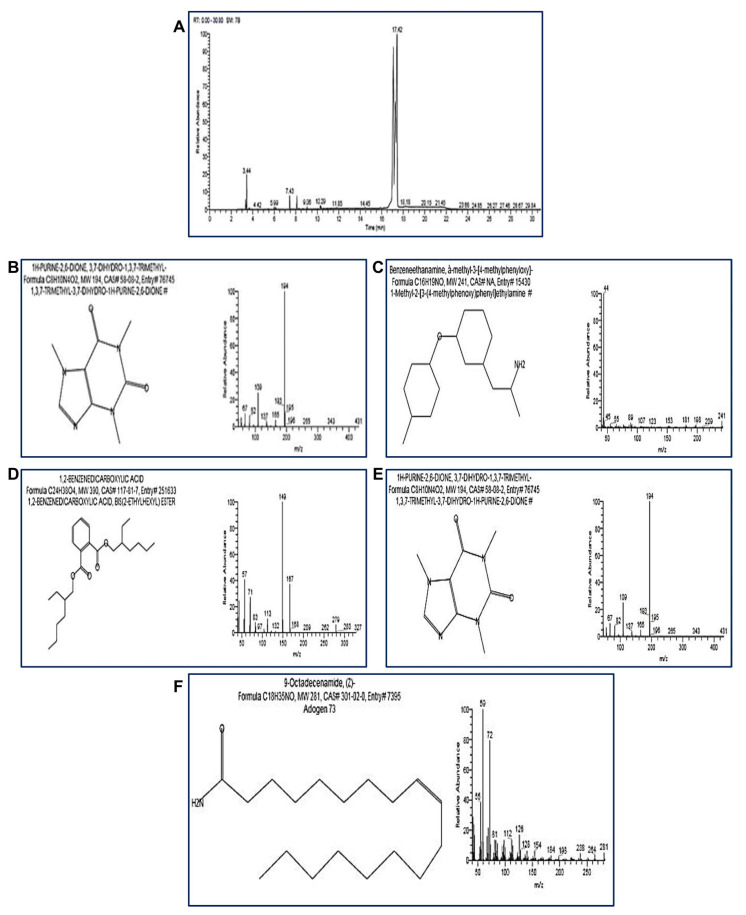
GC–MS analysis of extracted soil sample containing cypermethrin (2500 mg kg^−1^). Panel (**A**) is control (extract from soil containing cypermethrin without bacterial inoculation); panels (**B**–**F**) are extracted from soil samples containing cypermethrin and inoculated by *L. cresolivuorans* HIS7.

**Figure 8 plants-10-01903-f008:**
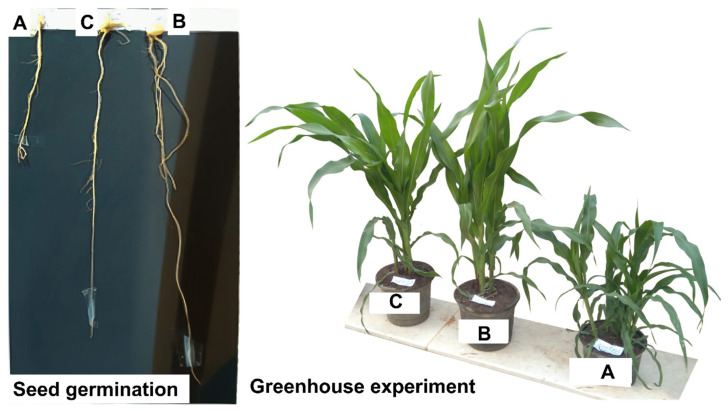
Toxicity assessment of pyrethroid insecticide cypermethrin before and after bacterial degradation compared with distilled H_2_O on *Zea mays* plant. Panel (**A**) cypermethrin before bacterial degradation; panel (**B**) distilled H_2_O; (**C**), biodegradable products of cypermethrin after bacterial degradation.

**Figure 9 plants-10-01903-f009:**
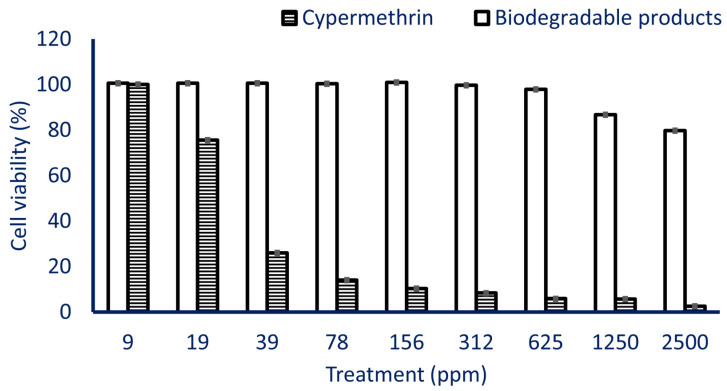
In vitro cytotoxic efficacy of cypermethrin before and after bacterial degradation on Vero cell line viability.

**Table 1 plants-10-01903-t001:** GC-MS analysis of extracted soil sample containing cypermethrin (2500 mg kg^−1^) in the absence (control) and presence of *L. cresolivuorans* strain HIS7.

Treatment	RT (min)	Percent (%)	Biodegradable Compounds	Molecular Formula	Molecular Weight
Control	17.08	31.51	Cypermethrin	C_22_H_19_Cl_2_NO_3_	415
* Lysinibacillus cresolivuorans * HIS7	4.43	0.20	Acetic acid (4-chloro-2-methylphenoxy)	C_9_H_9_ClO_3_	200
	9.20	4.37	1H-Purine-2,6-dione,3,7-dihydro-1,3,7trimethy	C_8_H_10_N_4_O_2_	194
	10.92	1.00	9-Octadecenamide	C_18_H_35_NO	281
	11.39	0.90	Benzene ethanamine, à-methyl-3-[4-methylphenyloxy]	C_16_H_19_NO	241
	11.80	1.87	1,2-Benzenedicarboxylicacid, 3-nitro	C_8_H_5_NO_6_	211

**Table 2 plants-10-01903-t002:** Effect of cypermethrin and its biodegradable end products on seed germination of *Zea mays* plant.

Treatment	Root Length (cm)	Root Biomass (mg)	
		Fresh Weight	Dry Weight
Positive control (distilled H_2_O)	29.0 ± 0.1	1715.4 ± 32.4	505.2 ± 23
Negative control (cypermethrin before bacterial degradation)	11.0 ± 0.06	862.5 ± 40.2	116.1 ± 14.7
Biodegradable products	25.0 ± 0.1	1655.5 ± 26.4	410.4 ± 12.4

**Table 3 plants-10-01903-t003:** Effect of cypermethrin and its biodegradable end products on growth performance of *Zea mays* plant.

Treatment	Root Length (cm)	Shoot Length (cm)	Growth Performance			
			Fresh Weight (g)		Dry Weight (g)	
			Root	Shoot	Root	Shoot
Positive control (distilled H_2_O)	14.0 ± 0.6	122.0 ± 0.7	3.6 ± 1.5	21.5 ± 2.3	1.9 ± 0.4	12.9 ± 2.2
Negative control (cypermethrin before bacterial degradation)	5.0 ± 0.4	88.0 ± 0.5	2.1 ± 1.0	13.2 ± 1.1	1.1 ± 0.6	5.3 ± 0.9
Biodegradable products	12.0 ± 0.6	115.0 ± 0.7	3.1 ± 1.1	19.9 ± 2.3	1.4 ± 0.2	11.9 ± 3.1

## Data Availability

The data presented in this study are available on request from the corresponding author.

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
