# Peer review of "Evaluate the Toxicity of Pyrethroid Insecticide Cypermethrin before and after Biodegradation by Lysinibacillus cresolivuorans Strain HIS7"

_plants, 2021, doi:10.3390/plants10091903_

Round 1

Reviewer 1 Report

I like the research work conducted by the authors on Biodegradation of Pyrethroid Insecticide Cypermethrin Using  Lysinibacillus cresolivuorans Strain HIS7 and Evaluate Their  Toxicity. However I have the following suggestion for the authors. 

  1. At the end of the introduction write keypoints whatever author did in this research pointwaie (1)...(2)..like that
  2. In title you are informing by strain HIS7 but in results 5 bacteria if you are giving information about all then give 16 SrDNA identification for all
  3. I am not agree with your tree redraw it. reduce the sequences,..Right now it is not looking good.
  4. Include in discussion all articles of the pyrethroids degradation some suggested are.doi.org/10.1016/j.chemosphere.2019.125507;  doi: 10.1007/s13205-016-0372-3;doi: 10.1007/s13205-016-0541-4
  5. statistical analysis and data is very clear and there is no more comments
  6. Please propose the degradation pathway with the respective strain on the basis of metabolites you observed
  7. my overall recommendation positive please revise for all the possible mistakes

Reviewer 2 Report

Ebrahim Saied et al. authored the manuscript titled "Biodegradation of Pyrethroid Insecticide Cypermethrin Using Lysinibacillus cresolivuorans Strain HIS7 and Evaluate Their Toxicity". The paper can not be accepted. English of the paper needs extensive editing before submission. The data looks odd. For detailed comments please see the attachment.

Reviewer 3 Report

The subject of the manuscript „Biodegradation of pyrethroid insecticide cypermethrin using
Lysinibacillus cresolivuorans Strain HIS7 and evaluate their toxicity” is consistent with the scope of the Journal. The work is concise and comprehensive. The title of the work corresponds to its content. The abstract, introduction and results and discussion have been written correctly, and the applied research methods do not raise any objections.

There are several shortcomings in the manuscript, however, they do not diminish its quality and can be quickly corrected.

For example:

1.The notation "OD" is written in the manuscript as "O.D." and "OD" - please standardize;

2. Line: 285, 302, 335 .. instead of “2500 mg Kg-1” should be “2500 mg kg-1”. It should be in accordance with the requirements of the journal in SI Units (International System of Units).

3. The formulas are incorrectly written (line: 498, 648) - question marks are visible.

4. Figure 7. The figure is good, but not very legible. Try to improve its quality.

5. Line 279: instead of “0,391” it should be “0.391”

6. Line 281: instead of “Figure 6. the biodegradation..” it should be “Figure 6. The biodegradation ..”

7. Line 339: instead of “2.7.1. seed germination of Zea mays” it should be “2.7.1. Seed germination of Zea mays"

Round 2

Reviewer 2 Report

I am OK with the revised version. But the first sentnece in the INTRODUCTION can be "Chemical insecticides, pesticides, and heavy metals are considered some of the main pollution sources for soil, groundwater, and other water ecosystems".

Reviewer 4 Report

The authors have perfectly attended to the aspects indicated in the first review. I consider tha the manuscript is suitable of publication be this form.